# Immunohistochemical Markers of the Epithelial-to-Mesenchymal Transition (EMT) Are Related to Extensive Lymph Nodal Spread, Peritoneal Dissemination, and Poor Prognosis in the Microsatellite-Stable Diffuse Histotype of Gastric Cancer

**DOI:** 10.3390/cancers14246023

**Published:** 2022-12-07

**Authors:** Daniele Marrelli, Luigi Marano, Maria Raffaella Ambrosio, Ludovico Carbone, Luigi Spagnoli, Roberto Petrioli, Alessandra Ongaro, Stefania Piccioni, Daniele Fusario, Franco Roviello

**Affiliations:** 1Department of Medicine, Surgery and Neurosciences, Unit of General Surgery and Surgical Oncology, University of Siena, Strade Delle Scotte 14, 53100 Siena, Italy; 2Pathology Unit, Azienda Sanitaria Toscana Nord-Ovest, Via Cocchi 1, 56121 Pisa, Italy; 3Department of Medicine, Surgery and Neurosciences, Unit of Medical Oncology, University of Siena, Strade Delle Scotte 14, 53100 Siena, Italy

**Keywords:** gastric cancer, diffuse histotype, epithelial-to-mesenchymal transition, microsatellite-stable tumors, immunohistochemistry, prognosis, outcome

## Abstract

**Simple Summary:**

The epithelial-to-mesenchymal transition (EMT) is a biological process in which epithelial cells transform into mesenchymal-like cells that are capable of invasion, migration, and metastasis. EMT-positive diffuse gastric cancer shows a strong association with extensive lymph nodal metastases, advanced pTNM stage, peritoneal dissemination, chemo-resistance, and poor prognosis. E-cadherin, CD44, and ZEB-1 are cheap immunohistochemical markers of the EMT phenotype. Within the Lauren diffuse histotype, EMT status identifies two different phenotypes (EMT− and EMT+) with distinct clinico-pathological and prognostic characteristics.

**Abstract:**

Background: Although the prognostic value of the epithelial-to-mesenchymal transition (EMT) in gastric cancer has been reported in several studies, the strong association with the diffuse type may represent a confounding factor. Our aim is to investigate potential correlations among EMT status, tumor advancement, and prognosis in diffuse gastric cancer. Methods: Between 1997 and 2012, 84 patients with microsatellite-stable (MSS) diffuse-type tumors underwent surgery. The EMT phenotype was assessed with the E-cadherin, CD44, and zinc finger E-box binding homeobox 1 (ZEB-1) immunohistochemical markers. Results: Forty-five out of 84 cases (54%) were EMT-positive; more advanced nodal status (*p* = 0.010), pTNM stage (*p* = 0.032), and vascular invasion (*p* = 0.037) were observed in this group. The median numbers of positive nodes (13 vs. 5) and involved nodal stations (4 vs. 2) were higher in the EMT-positive group. The cancer-related survival time was 26 months in EMT-positive cases vs. 51 in negative cases, with five-year survival rates of 17% vs. 51%, respectively (*p* = 0.001). The EMT status had an impact on the prognosis of patients with <70 years, R0 resections, or treatment with adjuvant chemotherapy. Tumor relapses after surgery and peritoneal spread were significantly higher in the EMT-positive tumors. Conclusions: EMT status, when assessed through immunohistochemistry, identified an aggressive phenotype of MSS diffuse-type tumors with extensive lymph nodal spread, peritoneal dissemination, and worse long-term outcomes.

## 1. Introduction

Despite its decreasing incidence, gastric cancer (GC) remains one of the most common causes of death for neoplasms worldwide. Recent epidemiological trends have indicated a relative increase in the rate of the diffuse histotype at the expense of the intestinal type in the West. This could explain the lack of improvements in survival and the increasing rates of peritoneal carcinomatosis in recent periods [1,2]. The intestinal and diffuse types of GC, aside from their morphological features, show evident pathological and clinical differences. The former is more common in males and older patients, whereas the diffuse type usually affects younger patients. The risk of lymph node metastasis is significantly higher in the diffuse type, as it is for extra-regional and para-aortic nodes, and the propensity for peritoneal recurrence is relevant, especially when the serosa is involved [3,4]. The aggressive features and peculiar pattern of spread have caused the consideration of different and more specific treatment approaches for this subgroup of GC, such as super-extended lymphadenectomy and locoregional treatments for the prevention of peritoneal recurrence [4]. However, in several reports, the Lauren histotype did not show an independent prognostic value, and this indicated that relevant heterogeneity may exist in the context of diffuse-type tumors [5,6,7].

In the last years, two molecular classifications of GC (The Cancer Genome Atlas Program, TCGA, and Asian Cancer Research Group, ACRG) have been introduced, and extensive research is now ongoing to explore potential clinical applications of different molecular subgroups [8,9,10]. Both classifications showed a simple division of GC into four subgroups; the ACRG divided GC into microsatellite instability (MSI) and microsatellite-stable tumors (MSS) group, with subsets of MSS tumors for epithelial-to-mesenchymal transition (EMT), TP53-positive, and TP53-negative groups [9]. The EMT is a process in which epithelial cells transform into mesenchymal-like cells that are capable of migration, invasion, and metastasis [11]. This process is associated with the downregulation of epithelial markers, such as E-cadherin, upregulation of mesenchymal markers, and involvement of the zinc finger E-box binding homeobox 1 (ZEB-1), a transcriptional repressor of E-cadherin expression and EMT promoter [12,13,14]. The stem-cell marker CD44, a non-kinase cell-surface transmembrane glycoprotein, is also a recognized immunohistochemical marker of the EMT phenotype [12].

The clinical significance of the EMT has been investigated in several neoplasms, and it has been found to be associated with advanced stages and poor prognosis, including in GC [9,12,15]. However, a strong association between EMT status and Lauren histology has been found, as more than 80% of EMT-positive tumors are of the diffuse histotype [9]. Genomic analyses showed that the signaling pathways of the EMT identify the so-called mesenchymal type of GC, and most of such cases are of diffuse histology [16]. Interestingly, poorly cohesive GC also showed a worse prognosis when associated with EMT signature [17]. To date, it is unclear if the association with diffuse histology acts as a confounding factor, or, alternatively, if EMT status could identify a more aggressive phenotype in the context of the diffuse-type group.

The aim of the present retrospective study was to evaluate the potential impact of EMT status on the clinical–pathological characteristics and survival of patients with MSS diffuse-type tumors treated with gastrectomy and lymphadenectomy, with a long follow-up period.

## 2. Materials and Methods

### 2.1. Patients

Patients with primary GC who submitted to surgical resection at the Department of General Surgery and Surgical Oncology, University of Siena, Italy, between 1997 and 2012 were considered. For the present study, the following selection criteria were adopted: (1) availability of a tumor specimen in the tissue bank; (2) the patients submitted to up-front resective surgery, and neo-adjuvant or non-resected cases were excluded; (3) MSS tumors, with the exclusion of MSI tumors; (4) diffuse histotype according to Lauren classification, with the exclusion of intestinal or mixed types. All clinical and pathological data were retrospectively retrieved from a prospectively collected database, as previously reported [1,18]. Informed consent for the use of clinical data and for molecular analysis was obtained from all patients at the time of hospital recovery. All clinical, pathological, and molecular procedures or analyses were performed in accordance with the institutional review committee of University of Siena.

### 2.2. Surgical Approach, Histopathology, and Staging

After routine clinical staging with a CT scan, patients with potentially resectable GC were addressed to surgery. The aim of the surgical procedure was complete resection of the tumor with negative resection margins and extended (D2, D2plus) lymphadenectomy. For tumors located in the lower and middle third, a subtotal gastrectomy was performed, maintaining a proximal margin of at least 5 cm from the tumor; in all other cases, a total gastrectomy was performed. After surgery, cases were classified as R1/R2 when microscopic (peritoneal cytology, resection margins) or macroscopic residual tumor was present.

Immediately after surgical resection, lymph node mapping was performed in the fresh specimen according to the Japanese classification, as previously reported. The number of removed and involved lymph nodes for each station was recorded. Lymph node stations were also divided into perigastric (stations 1 to 7), second level (stations 8 to 12) and third level stations (stations 13 to 16) according to the Japanese guidelines [19]. For TNM staging, all cases were re-classified according to the 8th edition of the AJCC/UICC TNM classification. All other pathological factors were classified as previously reported in other studies by our group and according to the last WHO classification of digestive system tumors [18].

### 2.3. Microsatellite Analysis and Immunohistochemistry (IHC)

Frozen samples of tumor tissue were retrieved for each patient in our biological bank. Microsatellite analysis was evaluated by using 5 quasi monomorphic mononucleotide repeats, namely, BAT-26, BAT-25, NR -24, NR-21, and NR-27, as previously reported in detail [18]. We considered a tumor as having MSI whenever 2 or more markers showed instability on 5 loci (MSI-H).

For IHC, all procedures were automatically carried out on representative 2 µm thick paraffin sections from each case by Benchmark Ultra (Ventana, Monza, Italy) by using extended antigen retrieval and with DAB as chromogen. The following antibodies and dilutions were used: E-cadherin (NCH-38, #M3612 Dako Denmark A/S, 1:100), CD44 (ab6124, #F10-44-2, Abcam, 1:100), and ZEB-1 (2C1a, #ab50887 Abcam, 1:10). Counterstain with Mayer’s Hematoxylin was applied for 2 min to each section. For each case, the IHC expression in the malignant cells was compared to the immunoreaction in normal epithelial and/or stromal cells (positive control). All markers were scored independently by two observers who were blinded to clinical data and to each other.

For E-cadherin, a three-tiered score of positivity was applied by evaluating the percentage of stained tumor cells (0–10% = none; 11–89% = 1; >90% = 2), whereas for CD44 and ZEB-1, a two-tiered score was used (presence: immunostaining in 10% or more of the epithelial tumor cells; absence: immunostaining in less than 10%).

Different patterns of expression (membranous, cytoplasmic, nuclear) were considered for different markers. The expression of E-cadherin was classified as follows: (1) “normal/positive”: membranous immunostaining in more than 10% of the epithelial tumor cells; (2) “lost/negative”: absence of immunostaining or positivity in less than 10% of the epithelial tumor cells; (3) “abnormal”: cytoplasmic or heterogeneous expression (between 10 and 90% of the tumoral cells were immunopositive at the membrane and cytoplasmatic level). Altered expression of E-cadherin was defined as either complete loss of membrane staining or an aberrant cytoplasmic staining pattern [20,21]. Membranous expression for CD44 (>10%) and nuclear expression for ZEB-1 (>10%) were considered for the assessment of marker positivity [22,23]. Samples with differences in assessments between the two investigators were re-evaluated, and a consensus decision was made. Because the staining patterns sometimes varied within the same tumor, particularly when the differentiation status varied, the final score was based on the dominant pattern, and the highest value was used for the statistical analysis.

We considered all samples in which E-cadherin was aberrant (negative/abnormal) and at least one of the EMT-related markers (CD44 or ZEB-1) was positive as showing an EMT phenotype [12,24] (Figure 1 and Figure 2).

### 2.4. Additional Treatments and Follow-Up

Adjuvant chemotherapy was proposed after surgery according to the tumor stage, R-status, and the patient’s general conditions and age. Then, patients were subjected to regular outpatient follow-up examinations according to a standard protocol [5]. The follow-up was closed in December 2021. Follow-up data were available for all patients under study. The median follow-up period was 30 months (range: 1–240) for the entire series and 96 months (range: 16–240) for patients who were still alive at the last check-up or had non-cancer-related deaths.

### 2.5. Statistical Analysis

Statistical analysis was performed by using the SPSS statistical software (21.0). Numerical variables were expressed as the mean and standard deviation (SD) when normally distributed and as the median and interquartile range (IQR) if not normally distributed. Statistical associations between clinicopathological characteristics and EMT status were assessed with a χ^2^ test for categorical variables and with a *t*-test or non-parametric tests (Mann–Whitney) for continuous variables. The linear regression (R^2^) was used to explore the correlations between two linear variables. Survival curves were estimated by using the Kaplan–Meier method. Death due to cancer (cancer-specific survival) was considered as the endpoint; as such, deaths due to causes other than gastric cancer were considered censored cases at the time of death. Survival curves were compared by using the log-rank test. A *p*-value < 0.05 was considered statistically significant.

## 3. Results

### 3.1. Patients and Treatment

A total of 84 patients (males, 49; median age: 64, range: 25–90) fulfilled the selection criteria and were included in the analysis. Most tumors (73%) were in the two distal thirds. A subtotal gastrectomy was performed in 45 patients (54%), and a total gastrectomy was performed in 39 patients. The median number of lymph nodes removed was 41 (IQR: 31–56). An R0 resection was obtained in 56 patients (67%). At histology, most cases were classified in advanced stages; the serosa was involved in 62 cases (74%), and only 11 patients had negative lymph nodes (13%). A median number of nine positive nodes (IQR: 2–22) was found, leading to an advanced nodal status in most patients. Twenty-five patients (30%) were classified as M1 due to macroscopic metastases, extra-regional positive nodes, or positive peritoneal washing.

### 3.2. Correlations between EMT Status and Clinical-Pathological Variables

After the IHC analysis, 45 cases (54%) were classified as EMT-positive, and 39 (46%) were classified as EMT-negative. In Table 1, the correlations between EMT status and clinical-pathological variables are reported. No significant differences were found according to age, gender, tumor location, or WHO histotype between the two groups. Regarding tumor advancement, although the pT distributions were similar, more advanced N status (*p*-value 0.010) and TNM stages (*p*-value 0.032) were observed in EMT-positive cases. Vascular invasion was also more frequent in these groups (*p*-value 0.037), whereas no significant differences were found in the number of removed lymph nodes, R status, or adjuvant therapy after surgery.

The extended lymphadenectomy and routinary lymph node mapping performed in our center allowed a more detailed analysis of the lymph node status according to EMT status. As reported in Figure 3a, the median number of positive nodes was significantly higher in the EMT-positive group than in the negative group: 13 (IQR: 6–24) vs. 5 (IQR: 1–17), respectively (*p*-value 0.010). Furthermore, the median number of metastatic nodal stations was also higher in the former (4, IQR: 2–6, vs. 2, IQR: 1–4; *p*-value 0.022) (Figure 3b). A significant difference in the involvement of lymph node stations was observed for perigastric lymph nodes (stations 1 to 7) according to the JGCA classification, whereas the involvement of second- and third-level nodal stations was similar in the two groups under study. In the EMT-positive group, the linear correlation between the number of removed and positive lymph nodes showed a higher and significant R^2^ with respect to the negative cases (Figure 3c).

### 3.3. Survival Analysis

A significant difference in cancer-specific survival was observed between the two groups under study (Figure 4). In the EMT-positive cases, the median survival time was 26 months with a 17% five-year survival rate vs. a median survival time of 51 months and 51% five-year survival rate for the EMT-negative group (*p*-value 0.001). Survival data were also stratified according to the main clinical–pathological variables (Table 2). A significant impact of the EMT status on the prognosis was observed in patients younger than 70 years and in tumors located in the middle/lower third; EMT-positive status also affected survival in poorly cohesive carcinomas independently of the signet-ring-cell-associated phenotype. Regarding treatment, a significant prognostic value of EMT status was observed in patients submitted to R0 resection and in the group treated with adjuvant chemotherapy.

A notable difference was also observed in tumors that did not involve the serosa, with an 89% five-year survival probability for EMT-negative cases; on the other hand, prognosis was very poor in EMT-positive tumors involving the serosa (Figure 5a,b).

A more in-depth analysis was also performed according to age. EMT status strongly affected survival in patients younger than 70 years; on the contrary, in aged patients, the survival curves overlapped with low cancer-specific survival rates independently of EMT status (Figure 5c,d).

The interaction between age and the clinical-pathological variables under study revealed that, of all considered factors, only the use of adjuvant chemotherapy differed significantly according to the age groups (93.2% in the younger group vs. 48% in the aged patients, *p*-value < 0.001). All other factors, including tumor stage and WHO histotype, were not significantly different (Marrelli D, Siena University, Italy; personal data).

The pattern of recurrence after surgery was also investigated (Table 3). This analysis showed that the recurrence rate and peritoneal dissemination were much higher in the EMT-positive group (33.3% vs. 12.8% of negative cases).

## 4. Discussion

The new molecular classifications of GC are recent and important steps in the management of this complex disease, and the flourishing of the literature is ongoing in order to clarify their potential clinical implications in clinical practice [4]. However, few clinical-oriented studies with detailed surgical data have been reported in the literature, and short follow-up periods are generally available for survival analyses.

The MSI group is the most studied in the literature, and it has been demonstrated to be associated with older age, less advanced stage, limited lymph node involvement, and better prognosis [8,9,15,18,25]. The EMT group of the ACRG classification is also very interesting from a clinical point of view [9]. The EMT is a process in which epithelial cells are transformed into cells with mesenchymal phenotypes, loss of cellular adhesion, enhanced invasive properties, and abundant stroma. These alterations, together with microenvironment remodeling, facilitate the aggressiveness, metastasis, and chemo-resistance of GC [11,14].

Some reports have suggested that the EMT phenotype is correlated with younger age, diffuse/poorly differentiated histology, advanced TNM stage, and worse prognosis of GC [9,11,12,15,16,26,27,28,29]. The strong association with the diffuse histotype or undifferentiated histology may act as a confounding factor, and it is unclear if this association could explain the clinical behavior of EMT-positive tumors [9,12,28,29]. For this reason, we focused our analysis on MSS diffuse-type tumors. A separate analysis of molecular features according to the Lauren histotype was also suggested by other authors [20]. To our knowledge, this is the first report to identify EMT status as a strong marker of tumor progression and poor prognosis in MSS diffuse-histotype GC patients with a long follow-up time.

In the ACRG study, more than 80% of the EMT cases were diffuse, but less than 50% of the diffuse-type cases belonged to the EMT subgroup [9]. The percentage of diffuse-type cases showing an EMT phenotype in our series (54%) was like the rate found in the ACRG study, as well as other reports [9,20,29]. In several studies, the Lauren classification was not used, and this did not allow a comparison with the present series [12,15].

Most clinical and pathological variables analyzed in the present study were similar according to the EMT status; however, the EMT-positive group showed a strong association with extensive lymph nodal metastases. The extended lymphadenectomy (median number of 41 removed lymph nodes) and the routinary mapping of lymph node stations allowed an accurate topographic analysis of nodal spread according to the molecular type. The median numbers of positive lymph nodes and involved nodal stations were notably higher in the EMT-positive group; only three patients (6.7%) had negative lymph nodes, and 71% of cases were pN3. Although it is well known that the Lauren histotype affects the incidence, number, and distant location of lymph node metastases in GC, the EMT status could characterize a phenotype with more extensive nodal spread. The strong linear correlation between the numbers of removed and positive nodes is indicative of the biological aggressiveness of EMT-positive tumors and their propensity to invade multiple nodal chains (Figure 3). However, according to the present data, this seems to happen, above all, for perigastric nodes, whereas the overall involvement rates of second- and third-level nodal stations were similar in the two groups under study. As such, from a surgical point of view, the extent of lymphadenectomy should not change according to EMT status in the diffuse histotype of GC.

Another important finding of the present study is the prognostic impact of EMT status in diffuse-type tumors. Most EMT-positive patients died due to GC during follow-up, with a median survival of 26 months vs. 51 months for negative cases, and the prognosis was particularly poor in advanced nodal stages and neoplasms involving the serosa. A greater propensity of EMT-positive tumors for peritoneal dissemination was also observed; on the contrary, the long-term survival and peritoneal spread of EMT-negative cases resemble the characteristics of the intestinal type. As such, the EMT phenotype may play a crucial role in the clinical correlation between diffuse-type and peritoneal carcinomatosis. Although the diffuse type has been proposed to be included in a unique pathological/molecular category, the present data indicate that a phenotype with more aggressive clinical behavior can be identified [30]. Interestingly, EMT status was equally predictive of worse survival in both signet- and non-signet poorly cohesive carcinomas of the WHO classification, which are emerging prognostic factors in GC [31,32]. These aspects should be better elucidated in future specific studies.

The present study included patients who were treated in the pre-neoadjuvant era and then submitted to up-front surgery, even for locally advanced forms. Although this may be considered a limitation, the analysis of the correlations among EMT status, tumor advancement, and lymph node spread is reliable in neoplasms that are not submitted to neo-adjuvant treatments. These results indicate that patients with EMT-positive diffuse-type tumors have a poor probability of surviving in the long term when submitted to up-front surgery. The prognostic value of EMT status in patients submitted to neo-adjuvant chemotherapy remains unclear and should be analyzed in future studies.

In our series, a favorable prognostic impact of negative EMT status was observed in patients submitted to adjuvant chemotherapy. Interestingly, this phenotype was associated with better prognosis at younger ages, whereas the long-term survival of aged patients was generally poor in both groups, independently from EMT status. It is important to note that almost all patients (93%) in the younger group were treated with adjuvant chemotherapy, in comparison with 48% of the aged patients. These data may fit with the hypothesis that EMT status could affect the response to chemotherapy in GC [11,33], but specific studies should be designed in the future.

The limitations of the present study include its retrospective design, its relatively small sample size, and the methods used for EMT evaluation. Although IHC may represent a limitation when compared with genomic analysis, this method is strongly encouraged for the molecular characterization of GC due to its limited costs. When confirmed in further studies, it could be an important gain for potential applications in a clinical setting [20,27,28].

The present findings need confirmation in external validation studies, and correlation analyses that are performed on endoscopic biopsies are necessary before drawing clinical and therapeutic implications. Potential changes in EMT markers induced by chemotherapy should also be considered. However, the present results indicate that the analysis of EMT status could expand the molecular characterization of GC. Preoperative biopsy of GC should focus on the assessment of MSI, EBV, and EMT phenotypes. The evaluation of this panel of molecular markers could entail indications for neo-adjuvant therapy, up-front surgery, and, in EMT-positive cases, a multimodal approach that includes hyperthermic intraperitoneal chemotherapy (HIPEC) or pressurized intraperitoneal aerosol chemotherapy (PIPAC), in consideration of the greater propensity for peritoneal spread [34]. Potential therapies with EMT inhibitors are under evaluation in clinical studies, and this could be another important step in a specific therapeutic approach to diffuse GC [35,36].

## 5. Conclusions

The results of the present study indicate that EMT status could be able to be used to identify a phenotype of diffuse-type tumors with a more advanced stage, extensive lymph nodal spread, peritoneal dissemination, and poor prognosis. This could have important scientific and clinical implications in the context of a tailored therapeutic approach to GC.

## Figures and Tables

**Figure 1 cancers-14-06023-f001:**
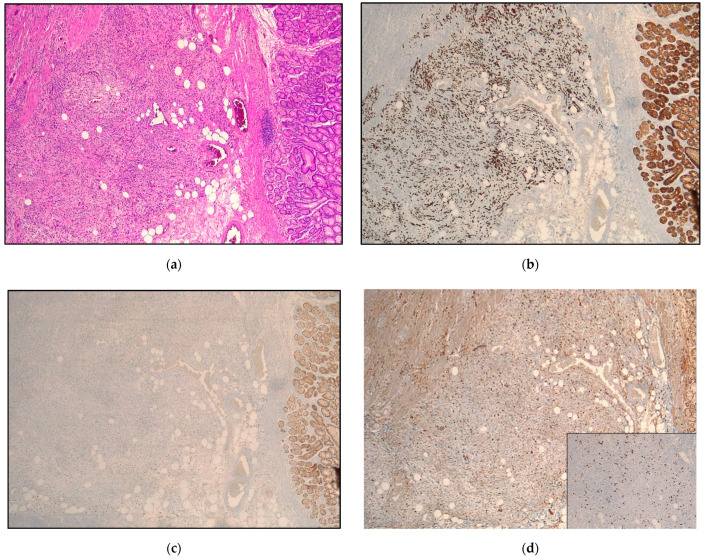
Representative samples of positivity for epithelial-to-mesenchymal transition markers: (**a**) A poorly cohesive signet ring cell phenotype (acc. to WHO) or diffuse type (acc. to Lauren) infiltrating the muscular layer is presented; (**b**) a 8/18 cytokeratin stain confirming the epithelial origin of the neoplastic proliferation is shown; (**c**) E-cadherin negativity of neoplastic cells is depicted; the positive control is represented by the normal gastric epithelium on the right side expressing E-cadherin; (**d**) neoplastic cells show membranous positivity for CD44 and nuclear expression of Zeb-1 (inset, bottom right) in more than 10% of cells. (**a**) Hematoxylin and eosin; (**b**) CK8/18 stain; (**c**) E-cadherin stain; (**d**) CD44 stain; inset on the bottom right, Zeb-1 stain. Original magnification (OM): (**a**–**d**), 4×; (**d**), inset, 10×.

**Figure 2 cancers-14-06023-f002:**
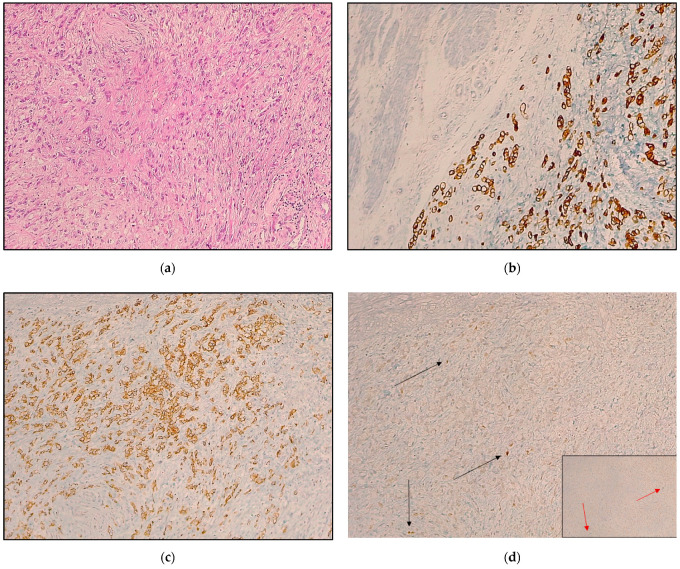
Representative sample of negativity for epithelial-to-mesenchymal transition markers: (**a**) A poorly cohesive other cell type (WHO) or diffuse type (Lauren) is presented; (**b**) a 8/18 cytokeratin stain confirming the epithelial origin of the neoplastic proliferation is shown; (**c**) E-cadherin positivity of neoplastic cells is depicted; (**d**) neoplastic cells do not show positivity for CD44 and Zeb-1 (inset, bottom right); the positive control is represented by lymphocytes (arrows). (**a**) Hematoxylin and eosin; (**b**) CK8/18 stain; (**c**) E-cadherin stain; (**d**) CD44 stain; inset on the bottom right, Zeb-1 stain. Original magnification (OM): (**a**–**d**), 4×; (**d**), inset, 10×.

**Figure 3 cancers-14-06023-f003:**
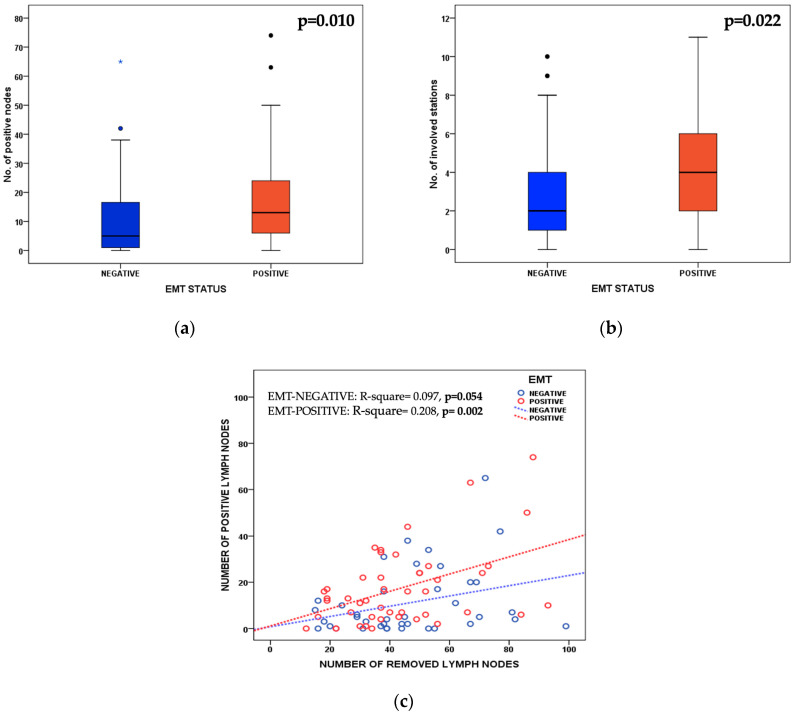
(**a**) Box-plot representation of the correlation between EMT status and the number of positive lymph nodes (Mann–Whitney U-test). The box represents the interquartile range (IQR), which contains 50% of the values; the lines extending from the box indicate the highest and lowest non-outlier values, while the line across the box indicates the median value. Circles represent “mild” outliers, i.e., values between 1.5 and 3.0 times the IQR; asterisks represent “extreme” outliers, i.e., values more than 3.0 times the IQR. (**b**) Box-plot representation of the correlation between EMT status and the number of metastatic lymph nodal stations classified according to the JGCA guidelines (Mann–Whitney U-test). A median number of 10 lymph node stations for patients were removed (range 6 to 15); a total number of 924 lymph node stations were analyzed. (**c**) Linear correlation between the number of removed and positive lymph nodes in EMT-positive and EMT-negative cases.

**Figure 4 cancers-14-06023-f004:**
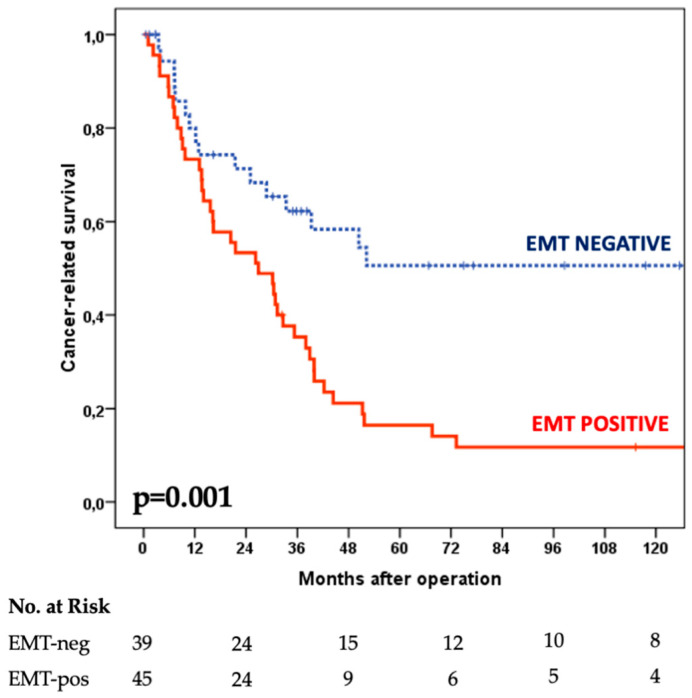
Cancer-specific survival according to EMT status in MSS diffuse-type tumors; the difference is statistically significant (Log-rank test).

**Figure 5 cancers-14-06023-f005:**
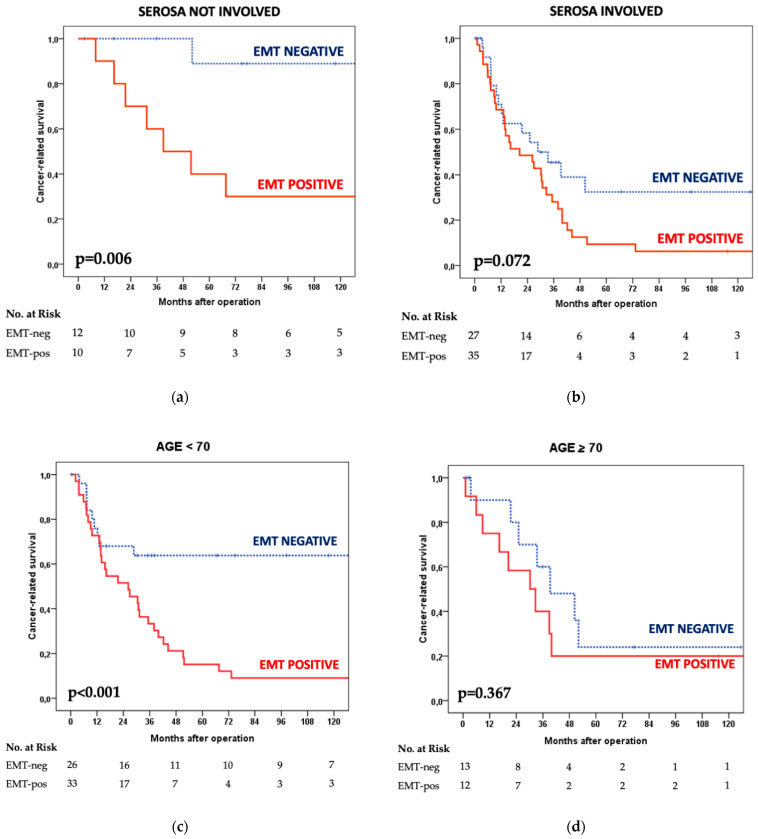
Cancer-specific survival according to EMT status: (**a**) MSS diffuse-type tumors that did not involve the serosa and (**b**) MSS diffuse-type tumors that involved the serosa; stratified for age groups: (**c**) <70 years old and (**d**) ≥70 years old (Log-rank test).

**Table 1 cancers-14-06023-t001:** Association between EMT status and clinical–pathological variables in MSS diffuse-type gastric cancer.

Characteristics	EMT-NEG N = 39	EMT-POS N = 45	*p*-Value
Age: median (IQR)	65 (54–72)	61 (52–70)	0.365 *
Gender			0.222 **
Male	20 (51.3%)	29 (64.4%)	
Female	19 (48.7%)	16 (35.6%)	
Tumor location			0.827 **
Upper third	5 (12.8%)	8 (17.8%)	
Middle third	9 (23.1%)	11 (24.4%)	
Lower third	21 (53.8%)	20 (44.4%)	
Linitis plastica	4 (10.3%)	6 (13.3%)	
WHO histotype			0.739 **
Signet-ring cell	29 (74.4%)	32 (71.1%)	
Other poorly cohesive	10 (25.6%)	13 (28.9%)	
Depth of tumor invasion			0.249 **
pT1	3 (7.7%)	1 (2.2%)	
pT2	4 (10.3%)	1 (2.2%)	
pT3	5 (12.8%)	8 (17.8%)	
pT4	27 (69.2%)	35 (77.8%)	
Lymph node involvement			0.010 **
pN0	8 (20.5%)	3 (6.7%)	
pN1	9 (23.1%)	3 (6.7%)	
pN2	9 (23.1%)	7 (15.6%)	
pN3a	4 (10.3%)	11 (24.4%)	
pN3b	9 (23.1%)	21 (46.7%)	
Presence of metastasis			0.442 **
M0	29 (74.4%)	30 (66.7%)	
M1	10 (25.6%)	15 (33.3%)	
TNM Stage			0.032 **
I–II	13 (33.3%)	5 (11.1%)	
IIIA–IIIB	13 (33.3%)	14 (31.1%)	
IIIC	3 (7.7%)	11 (24.4%)	
IV	10 (25.6%)	15 (33.3%)	
Lymphatic invasion			0.660 **
Absent	13 (33.3%)	13 (28.9%)	
Present	26 (66.7%)	32 (71.1%)	
Vascular invasion			0.037 **
Absent	18 (46.2%)	11 (24.4%)	
Present	21 (53.8%)	34 (75.6%)	
Perineural invasion			0.437 **
Absent	18 (46.2%)	17 (37.8%)	
Present	21 (53.8%)	28 (62.2%)	
Peritoneal cytology			0.433 **
Negative	12 (30.8%)	15 (33.3%)	
Positive	5 (12.8%)	10 (22.2%)	
N.A.	22 (56.4%)	20 (44.4%)	
R status			0.353 **
R0	28 (71.8%)	28 (62.2%)	
R1–R2	11 (28.2%)	17 (37.8%)	
Removed lymph nodes (median, IQR)	44 (31–62)	38 (31–53)	0.422 *
Lymph node stations involved			
(JGCA classification)			
Stations #1 to 7	29 (74.4%)	42 (93.3%)	0.036 **
Stations #8 to 12	16 (41.0%)	21 (46.7%)	0.765 **
Stations #13 to 16	6 (15.4%)	7 (15.6%)	1.000 **
Adjuvant chemotherapy			0.091 **
No	11 (28.2%)	6 (13.3%)	
Yes	28 (71.8%)	39 (86.7%)	

N.A.: not available; IQR: interquartile range; JGCA: Japanese Gastric Cancer Association; * Mann–Whitney U-test; ** two-tailed chi-square.

**Table 2 cancers-14-06023-t002:** Five-year cancer-specific survival probability according to EMT status, stratified for the main clinical-pathological factors.

Characteristics	No. of Cases	EMT-NEG 5-Year Survival (±SE)	EMT-POS 5-Year Survival (±SE)	*p*-Value *
Gender				
Male	49	53% ± 13	17% ± 7	0.025
Female	35	48% ± 13	14% ± 9	0.025
Age				
<70	59	63% ± 10	15% ± 6	0.001
>70	25	24% ± 14	20% ± 12	0.367
Tumor location				
Upper third	13	0	0	0.928
Middle third	20	78% ± 14	11% ± 10	0.004
Lower third	41	52% ± 13	25% ± 10	0.028
Linitis plastica	10 **	N.A.	N.A.	N.A.
WHO histotype				
Signet-ring cell	61	51% ± 10	17% ± 7	0.036
Other poorly cohesive	23	44% ± 23	15% ± 10	0.012
Depth of tumor invasion				
pT1–T3	22	89% ± 10	40% ± 16	0.006
pT4	62	33% ± 11	9% ± 5	0.072
Lymph node involvement				
pN0	11	50% ± 30	33% ± 27	0.514
pN1–N2	28	73% ± 11	30% ± 14	0.854
pN3a	15	33% ± 27	18% ± 12	0.470
pN3b	30	22% ± 14	0	0.967
TNM Stage				
I–II	18	79% ± 13	40% ± 22	0.087
IIIA–IIIB	27	56% ± 15	29% ± 12	0.138
IIIC–IV	39	25% ± 13	5% ± 4	0.844
R status				
R0	56	65% ± 10	25% ± 8	0.001
R1–R2	28	11% ± 10	0	0.603
Adjuvant chemotherapy				
No	17	40% ± 17	17% ± 15	0.137
Yes	67	56% ± 10	16% ± 6	0.006

SE: standard error; * log-rank test; ** analysis not performed due to the low number of cases. N.A.: not available.

**Table 3 cancers-14-06023-t003:** Tumor recurrence after surgery according to EMT status.

Recurrence	EMT-NEG N = 39	EMT-POS N = 45	*p*-Value *
			<0.001
No recurrence	23 (59.0%)	6 (13.3%)	
Local	4 (10.3%)	9 (20.0%)	
Liver	2 (5.1%)	3 (6.7%)	
Peritoneum	5 (12.8%)	15 (33.3%)	
Multiple sites	1 (2.6%)	4 (8.9%)	
Unknown site	4 (10.3%)	8 (17.8%)	

* Two-tailed chi-square.

## Data Availability

Not applicable.

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
