# Peer review of "Immunohistochemical Markers of the Epithelial-to-Mesenchymal Transition (EMT) Are Related to Extensive Lymph Nodal Spread, Peritoneal Dissemination, and Poor Prognosis in the Microsatellite-Stable Diffuse Histotype of Gastric Cancer"

_cancers, 2022, doi:10.3390/cancers14246023_

Round 1

Reviewer 1 Report

Although retrospective, the study is based on an intersting hypothesis and was well conducted. Methods are carefully described and the statistical approach is appropriate. Results are clearly reported and tha discussion is well-balanced leading to consistent conclusions.

The number of self-citations seems redundant; one or more of reference #5, 19, 31, 36-36 might be removed.

Author Response

Dear Reviewer, thank you very much for your interesting criticisms.

We removed those references and corrected the English language.

Sincerely.

Reviewer 2 Report

The study demonstrates that EMT phenotypic markers are related to extensive lymph nodal spread, peritoneal dissemination, and poor prognosis in microsatellite-stable diffuse-type gastric cancer.

Introduction may be revised to have clinical significance of EMT in terms of association between EMT status and Lauren histology more in detail in lines 74-79.

The meanings of circles and asterisks in Figure 1 a and b may be added in the legend. The term "involved station" may be explained more in detail in Figure 1b and the numbers of samples (N) may be added in the Figure legend.

Author Response

Dear Reviewer, thank you very much for your insightful remarks.

We detailed the association between EMT status and Lauren histology in Introduction section.

We added explanations and number of nodal stations in Figure 1 legend.

Sincerely.

Reviewer 3 Report

The authors have investigated the EMT status in patients with MSS and diffuse type gastric cancer. Their results indicate that the status of EMT is significantly correlated with lymph node spread, peritoneal dissemination, and long-term outcome. Focusing on restricted in subgroup of MSS and diffuse type gastric cancer to avoid confounding is reasonable. The study is well-conducted, and the manuscript is well written. I have following minor comments. Please provide the histological figures including immunohistochemistry of EMT markers. In abstract: Forty-five cases (54%), please correct it.

Author Response

Dear Reviewer, grateful for your comment.

We provided two figures (1 and 2) including immunohistochemistry of EMT markers.

We corrected the abstract.

Sincerely.